# Effects of Load Carriage on Postural Control and Spatiotemporal Gait Parameters during Level and Uphill Walking

**DOI:** 10.3390/s23020609

**Published:** 2023-01-05

**Authors:** Asimina Mexi, Ioannis Kafetzakis, Maria Korontzi, Dimitris Karagiannakis, Perikles Kalatzis, Dimitris Mandalidis

**Affiliations:** 1Sports Physical Therapy Laboratory, Department of Physical Education and Sports Science, School of Physical Education and Sports Science, National and Kapodistrian University of Athens, 17237 Athens, Greece; 2Section of Informatics 1st Vocational Lyceum of Vari, Directorate of Secondary Education of East Attica, Hellenic Ministry of Education and Religious Affairs, 16672 Athens, Greece

**Keywords:** front pack, backpack, EMG activity, trunk sway, gait analysis, erector spinae, rectus abdominis

## Abstract

Load carriage and uphill walking are conditions that either individually or in combination can compromise postural control and gait eliciting several musculoskeletal low back and lower limb injuries. The objectives of this study were to investigate postural control responses and spatiotemporal parameters of gait during level and uphill unloaded (UL), back-loaded (BL), and front-loaded (FL) walking. Postural control was assessed in 30 asymptomatic individuals by simultaneously recording (i) EMG activity of neck, thoracic and lumbar erector spinae, and rectus abdominis, (ii) projected 95% ellipse area as well as the anteroposterior and mediolateral trunk displacement, and (iii) spatiotemporal gait parameters (stride/step length and cadence). Measurements were performed during level (0%) and uphill (5, 10, and 15%) walking at a speed of 5 km h^−1^ without and with a suspended front pack or a backpack weighing 15% of each participant’s body weight. The results of our study showed that postural control, as indicated by increased erector spinae EMG activity and changes in spatiotemporal parameters of gait that manifested with decreased stride/step length and increased cadence, is compromised particularly during level and uphill FL walking as opposed to BL or UL walking, potentially increasing the risk of musculoskeletal and fall-related injuries.

## 1. Introduction

Postural control is defined as the act of maintaining, achieving or restoring a state of balance during any posture or activity and is under the constant regulation of the central nervous system (CNS) [1]. Consequently, any stimulus that is perceived as a threat to the balance of the body by the sensory afferent system is compensated or anticipated with appropriate motor responses to avoid its loss and eventually an injury and/or a fall [1]. Among the conditions that can significantly challenge postural control is load carriage. Load carriage is an integral part of everyday life for many civilians (e.g., students), professionals (e.g., law enforcement and rescue teams), and leisure athletes (e.g., hikers). In many cases, such packs either by design, manners, safety reasons, or working duties are carried in front of the trunk as occurs with day backpacks, babies, groceries, or items like those carried by workers in the construction, moving, or transportation-related industries. Placing a backpack at the front of the trunk, although it is not recommended for long periods, may also be temporarily helpful in relieving postural changes caused by carrying the backpack at the back of the trunk [2].

Occasionally, a front or a back load may also need to be carried uphill, further compromising postural control as the induced postural responses are expected to compensate for the forces generated by both the carried load and the increased slope of the ground [3,4,5]. Under certain circumstances these loads that often weigh up to 45 kg [6] may need to be carried on slopes that vary from 15° to 45° as required by army personnel for mountain operations [7]. However, for most citizens who choose to walk to and from an educational institution or workplace, to meet their daily needs or to engage in outdoor recreational activities (e.g., hiking), they are required to carry loads with much less weight on slopes that do not exceed 6° of inclination, i.e., ≈10% slope [8,9,10], unless they live in areas located on steep slopes [11].

The investigation of postural responses elicited by different load carrying modes has attracted the interest of many researchers whose aim was to identify the factors that contribute to musculoskeletal symptoms/injuries (e.g., back pain/strains) and slips/falls occurring in various populations (e.g., students [12], military personnel [13], workers [14], hikers [15]). It was thought that changes in the activity of the anti-gravity muscles of the trunk, the postural sway, and/or spatiotemporal gait parameters that occur in response to the application of a load may contribute to some extend in causing these injuries. In this context, postural control has been assessed by recording the EMG activity of trunk muscles, mainly of the spine, based on the fact that blood flow is restricted when static muscle effort increases up to 20% of maximum isometric voluntary contraction (MIVC) or may cease almost completely at about 60% MIVC, causing fatigue and eventually muscle deterioration and pain, especially if sustained for some time [16]. Using different research methods, equipment, loads, and test conditions, most studies to date have shown that the EMG activity of the lumbar paraspinal muscles either decreases or remains unaltered in response to load carriage at the back of the trunk during walking on a level surface, compared to unloaded walking [17,18,19,20] and unloaded upright standing [21]. In contrast, walking on a level surface with a load suspended or carried with the hands at the front of the trunk generates more thoracolumbar muscle activity than the standing trials [17,22]. Findings also showed that activation of superficial and deep posterior trunk muscles increased during uphill unloaded walking [23,24,25] and increased further, depending on the walking speed and surface slope, by carrying a back load [26].

Postural control has also been investigated in terms of the disturbances of body sway based on the general concept that the greater its magnitude or deviation the poorer the postural stability and possibly the greater the chance for falling [27]. Research thus far has shown that variables related to body sway such as ellipse area, path length, velocity of resultant, and/or mediolateral and anteroposterior displacement may increase when an individual carries a load at the front or the back of the trunk, but only a few having been collected during walking [28,29,30]. Load carriage-induced fall risk has been investigated also in terms of changes in the spatiotemporal gait characteristics but mainly during level walking. Based on a systematic review on the effect of backpack carriage on the biomechanics of level treadmill and overground gait, Liew et al. [31] reported that walking with a backpack is associated with reduced stride length and increased cadence while Simpkins et al. [32] in a similar study revealed that front-loaded walking can shorten the stride length, both in young and older adults, without compromising other gait variables. These findings imply that load carriage can compromise the biomechanics of walking, possibly leading to injuries.

Although more information is needed to understand the postural responses and gait adaptations induced by different modes of uphill load-carrying, research is still limited and related only to the study of uphill unloaded [23,24,25] and back-loaded walking [26]. By knowing the requirements needed to perform these activities, health care providers, exercise scientists, and ergonomists will be able to provide the populations involved with the necessary instruction to execute them in the most efficient and safe manner. The purpose of the study, therefore, was to investigate the effect of uphill front-loaded or back-loaded walking on postural control and spatiotemporal parameters of gait.

## 2. Materials and Methods

### 2.1. Study Sample

A total of 30 healthy and skeletally symmetric young adults (15 males and 15 females) aged 20–30 years without musculoskeletal pain syndromes (e.g., low back pain), injuries during the year prior to the study, and/or neurological problems participated in the study. Volunteers with musculoskeletal abnormalities and/or side-to side asymmetries such as excessive kyphosis, scoliosis (>5.0° of trunk rotation in the thoracic or lumbar spine during execution of the Adam’s test) [33], shoulder protraction asymmetry, leg length discrepancy (>0.5 cm), or pelvic asymmetry in the frontal or sagittal plane were excluded from the study (Table 1). The study commenced after the research proposal was approved by the Institution’s Research Bioethics Committee and all participants gave written consent after being informed about the experimental procedure and the aim of the study.

### 2.2. Musculoskeletal Assessment

The Flexicurve method was used to measure the thoracic curve by calculating the angular distance of an arc that corresponded to the contour of the thoracic spine [34]. Trunk rotation in the thoracic or lumbar spine during execution of the Adam’s test was measured with an inclinometer (scoliometer) [33]. Shoulder protraction was determined by measuring the horizontal distance between the anterior tip of the acromion and the wall using an adjustable double squared carpenter’s metal angle with each participant standing upright against the wall [35]. The length of the legs was measured from the anterior superior iliac spine (ASIS) to the lower tip of the medial malleolus of each limb using a common tape measure. The measurement was performed with each supine subject tested and with the hip and ankle joints in a neutral position for inward–outward rotation and pronation–supine, respectively [36]. Pelvic tilt and rotation were calculated based on the inverse trigonometric function of the tangent according to the formula θ = arctan (d/l) where d is the difference between the vertical or horizontal distances of the ASIS from the ground or from a fixed point in space, respectively, and l is the inter-ASIS distance, with each participant standing upright. All distances were measured using a common tape measure, and two laser-based technology distance meters (PLR 50, Robert Bosch GmbH, Leinfelden-Echterdingen, Germany) attached on a stadiometer specifically designed for pelvis position measurements [37]. All measurements were performed twice, and their average was used in the statistical analysis. The testing procedures have demonstrated high intraclass correlation coefficients (0.86–0.99) and therefore their reliability is considered acceptable for use in clinical practice [34,35,36,37].

### 2.3. Testing Procedure

The testing protocol required each participant to walk for 10 min at a speed of 5 km h^−1^ on a treadmill (RS-10 Treadmill SALTER, Spain), where the surface slope was increased every 2.5 min from 0% (level) to 5, 10, and 15% uphill. The walking speed with which participants performed the testing protocol in the present study was selected based on the speed that is usually adopted by healthy males and females aged 19–29 years when walking at a normal pace [38]. Level and uphill walking was performed without and by suspending a standard backpack (H: 49 cm × W: 27 cm) filled with books weighing 15% of each participant’s body weight at the front or the back of the trunk. The required mass consisted of books with a mass-to-thickness ratio of approximately 1.0 kg-to-1.5 cm. The straps of the backpack were adjusted so that the bottom of the backpack was located just above the iliac spines with the weight evenly and symmetrically distributed on both shoulders to avoid unilateral over-activation of the trunk muscles [39]. Paraspinal muscle EMG activity was acquired simultaneously with parameters related to trunk sway and gait during level and uphill unloaded (UL), back-loaded (BL), and front-loaded (FL) walking. This was achieved by starting the EMG activity recording device 10 s after starting the device recording the trunk sway and gait data (Figure 1). Data acquisition in all studied conditions was performed with the front pack and backpack in direct contact with the EMG electrodes and the IMU as the findings of a pilot study involving six subjects showed non-significant differences compared to the EMG and IMU data sets that were acquired without contact between the carried load and the sensors (Appendix A).

All measurements were performed with 5–7 min rest between walking conditions. Before the commencement of measurements, each subject performed a 10-min warm-up which consisted of a 5 min walk at a speed of 5 km h^−1^ and 5 min of stretching exercises for the muscles of the lower extremities. The potential effect of fatigue on the research data was prevented by randomly alternating the various testing conditions (slopes, loaded conditions).

### 2.4. EMG Activity Measurements

The EMG activity of the cervical, thoracic, and lumbar extensors of the trunk by means of the cervical, thoracic, and lumbar erector spinae (CES, TES, and LES) as well as of the rectus abdominis (RAB), a flexor of the trunk, was recorded using a bio signal recording system (BiopacMP 100, Aero Camino Goleta, CA, USA). The system consisted of a differential amplifier and four input channels, using self-adhesive monopolar Ag–AgCl electrodes 1.0 cm in diameter. Pairs of electrodes with an inter-electrode distance of 2 cm were placed on the right side of the participants’ body (i) 2 cm laterally from the C5 spinous process for the CES, (ii) 5 cm laterally from the spinous process of T9 for the TES, and (iii) 2 cm laterally from the L4 and L5 interspace for the LES in the direction of the muscle fibers. The electrodes for the trunk flexors were placed on the RAB, at the level of the ASIS, 2 cm laterally to the midline. The ground electrode was placed in the middle of the clavicle. Electrode placement sites on the skin surface were palpated with each individual in an upright position and cleaned with a cotton swab soaked in ethyl alcohol to maximize signal conductivity after the hair in the area were removed, if any.

The EMG activity of the muscles under investigation was recorded at a frequency of 1000 samples/s using a band pass filter (Band Pass FIR) frequency of 10–500 Hz. A factory preset 50 Hz notch filter, which matched the wall-power line frequency for most European countries, was used to reduce the noise from the interfering signals. The data was transmitted wired in real time to a computer, the connection of which to the recording system ensured the visualization, storage, and analysis of data using the system’s application software (AcqKnowledge, v. 3.9.1.6, Biopac Systems, Inc. Aero Camino Goleta, CA, USA).

Real time calculation of the root mean square (RMS) of the filtered signal using a window of 100 ms was utilized to analyze the activity of the muscles under investigation. The mean EMG activity corresponding to each step performed in the 120 intermediate seconds of walking at each slope and load-carrying mode divided by the number of steps was used in the statistical analysis. The steps included in the 120-s time interval were defined based on the time point of the step taken immediately after the completion of the 15th second of walking, as detected by the application software of the IMU/OCS system, until the time point of the step taken just before completing 120 of the 150-s (2 ½-min) of walk performed at each slope and load carrying mode (Figure 1). The EMG activity recorded at each step was calculated based on the step time provided by the system application software in milliseconds using a computer program written in Python 3.5, specifically developed for the purposes of this study.

The mean calculated RMS of the EMG activity produced during this time frame was expressed as a percentage of the mean RMS of the EMG activity produced during MIVC of each muscle (normalization). The MIVC for the neck, thoracic, and lumbar paravertebral muscles was induced by instructing each participant to extend the neck and the trunk against resistance applied by the examiner’s hands at the back of the head and the upper half of the trunk, respectively, while he/she was in the prone position, with the knees and hips fully extended and the hands blended between each other behind the neck. The MIVC of the RAB was measured during execution of a standard sit up (i.e., supine position with the knees bends at approximately 100° and feet secured on the treatment table, and hands blended behind the neck) against resistance applied manually by the examiner on each participant’s chest. The EMG activity of each muscle’s MIVC was recorded in two 5-s attempts and their average, based on the mean EMG activity produced in the intermediate 3 s, was considered in the normalization of the data.

### 2.5. Trunk Sway Measurements

Trunk sway was measured using an inertial sensor of 50 mm (width) × 70 mm (height) × 20 mm (thickness) and weight 35 g (Gyko, Microgate, Bolzano, Italy), which was synchronized with an optoelectric cell system (OCS) for gait analysis (Optogait, Microgate, Bolzano, Italy). The sensor, fixed on a special vest, was located at the back of each subject’s trunk, between the shoulder blades. The sensor consisted of a 3D accelerometer, a 3D gyroscope, and a 3D magnetometer that allow recordings of the trunk sway related parameters with a frequency of 1000 Hz. The signals from the inertial sensor were filtered by applying a low pass Butterworth filter with cutoff frequency of 30 Hz before being transmitted via Bluetooth technology to a computer where they were stored, displayed, and processed using the system’s application software (GykoRepower v. 1.2.2.0). Static calibration of the inertial sensor was performed with each subject standing still for few seconds at the beginning of each data collection to initialize the sensors fusion filter (i.e., the Mahony filter that provides as output the orientation of the sensor in the space). The trunk sway data provided by the software by means of the projected 95% ellipse area as well as the anteroposterior (AP) and mediolateral (ML) displacement of the trunk’s sway in each slope and loading condition was used in data analysis [40].

### 2.6. Measurement of Spatiotemporal Gait Parameters

Spatiotemporal gait parameters were measured via the OCS that consisted of optical sensors embedded into an emission and a receiving bar 100 cm (L) × 8 cm (W) in size. The optical sensors could detect any interruption in the light signal due to the presence of the participant’s feet within the recording area, with a recording frequency of 1000 Hz, as he/she walked between a pair of bars placed parallel to each other on the sides of the treadmill’s frame. The length of each step and stride and the cadence were recorded in each loading condition and slope.

### 2.7. Heart Rate and Perceived Exertion Measurements

Heart rate (HR) and the perceived exertion (PE) of physical activity were monitored during walking using a finger pulse oximeter (Fingertip Pulse Oximeter A1113, OEM, Tranås, Sweden) and the numbered 6 to 20 Borg’s 15-point rating scale of Perceived Exertion [41], respectively, 15 s before changing each slope.

### 2.8. Statistical Analysis

To achieve statistical significance with a = 0.05, 80% power, and effect size (f) = 0.25 (calculated based on a partial η^2^ = 0.06), an a priori power analysis using an online power analysis application (G*Power v. 3.1.9.2; FranzFaul, Universität Kiel, Kiel, Germany) was performed to determine an adequate sample size [42,43]. The results of the power analysis showed a total sample size of 30 individuals who ultimately participated in the current study.

Violations of statistical assumptions regarding the normality of the EMG data distribution, after being examined with the Shapiro–Wilk test and by visually observing the Q-Q and box plot graphs, necessitated a logarithmic transformation of the EMG raw signals. Logarithmic means and standard deviations were back transformed and presented as geometric means and a 95% confidence interval. A two-way repeated measures ANOVA was performed to detect any differences between the slope (0, 5, 10, and 15%) and the loading conditions (unloaded, backloaded, and front-loaded condition) under investigation for the EMG parameters, the trunk sway’s variables, and the spatiotemporal gait characteristics. Sphericity of the data was determined based on the Mauchly’s Test and where significant Greenhouse–Geisser correction was used. Pairwise comparisons were performed using the Bonferroni adjustment. The statistical analysis of the data was performed with SPSS 28.0 (IBM Corp, Armonk, NY, USA), while the significance level was set at the level of *p* ≤ 0.05.

## 3. Results

### 3.1. EMG Activity

The results of the present study showed significant main effects of slope (F = 32,668, *p* ≤ 0.001, partial η^2^ = 0.538), and loading condition (F = 16,197, *p* ≤ 0.001, partial η^2^ = 0.366), for the EMG activity of the CES. The slope-by-loading condition interaction was not significant. Post hoc tests revealed that the EMG activity of CES increased as the treadmill’s slope during UL, BL, or FL walking increased. Furthermore, EMG activity of CES during FL walking was greater at all slopes than the activity recorded during UL and BL walking (see pairwise comparisons in Figure 2).

Similarly, significant main effects of slope (F = 49,979, *p* ≤ 0.001, partial η^2^ = 0.641), and loading condition (F = 33,364, *p* ≤ 0.001, partial η^2^ = 0.544) were revealed for the EMG activity of the TES (see pairwise comparisons in Figure 3). The slope-by-loading condition interaction was not significant. The EMG activity of TES increased as the slope of the treadmill surface increased, and it was greater during uphill FL walking compared to UL and BL walking.

Statistical analysis yielded also significant main effect of slope (F = 83,590, *p* ≤ 0.001, η^2^ = 0.756) and loading condition (F = 19,847, *p* ≤ 0.001, partial η^2^ = 0.424) for the LES. A significant slope-by-loading condition interaction (F = 3460, *p* ≤ 0.05, partial η^2^ = 0.114), showed that although the EMG activity of LES increased as the slope of the treadmill’s surface increased, it was higher in the FL compared to the UL and BL condition with the latter two walking conditions demonstrating similar EMG activities (see pairwise comparisons in Figure 4).

A significant main effect of loading condition (F = 6872, *p* ≤ 0.01, partial η^2^ = 0.209) but not of slope was found for RAB. A significant slope-by-loading condition interaction (F = 3017, *p* ≤ 0.05, partial η^2^ = 0.104) showed that RAB demonstrated higher EMG activity in the BL and FL walking conditions compared to the UL walking condition with the EMG activity of RAB being progressively increased, albeit not statistically significant, as the slope increased only during uphill FL walking (see pairwise comparisons in Figure 5).

### 3.2. Trunk Sway

A significant main effect of slope (F = 23,389, *p* ≤ 0.001, partial η^2^ = 0.446), but not of loading condition, and a significant slope-by-loading condition interaction (F = 4487, *p* ≤ 0.05, η^2^ = 0.134) were found for the 95% ellipse area of trunk sway. Post hoc analysis revealed that the 95% ellipse area of trunk sway was significantly increased as the slope of the treadmill’s surface gradually increased in all walking conditions. The 95% ellipse area of trunk sway increased progressively more in the FL walking condition by being smaller at 0% slope and larger at 15% slope compared to the UL and BL walking conditions (see pairwise comparisons in Figure 6).

Statistical analysis revealed also significant main effect of slope for the AP trunk displacement (F = 119,483, *p* ≤ 0.001, partial η^2^ = 0.805), which was gradually increased with increased slope at all loading conditions (see pairwise comparisons in Figure 7). The main effect of the loading condition for the AP trunk displacement was significant (F = 3.326, *p* ≤ 0.05, partial η^2^ = 0.103) with post hoc analysis revealing significant differences only between FL and BL walking at 5% slope (*p* ≤ 0.05). The slope-by-loading condition interaction for the AL trunk displacement was not significant.

Significant main effects of slope (F = 5.191, *p* ≤ 0.01, partial η^2^ = 0.152) were obtained for ML trunk displacement but none of the pairwise comparisons were significant. The main effects for the loading condition and the slope-by-loading condition interaction were not significant (Figure 8).

### 3.3. Spatiotemporal Gait Parameters

Statistical analysis revealed significant main effects of loading condition (F = 51,657, *p* ≤ 0.001, partial η^2^ = 0.640) and a significant slope-by-load interaction (F = 6314, *p* ≤ 0.001, η^2^ = 0.179) for step length. The main effects of slope were not significant. Post hoc comparisons revealed longer step length during walking at 5, 10, and 15% slopes compared to level (0%) walking for the UL and FL carrying mode and lower step length during level and uphill FL walking compared to level and uphill UL and BL walking (see pairwise comparisons in Table 2).

Significant main effects of loading condition (F = 51,430, *p* ≤ 0.001, partial η^2^ = 0.639) and a significant slope-by-loading condition interaction (F = 6290, *p* ≤ 0.001, η^2^ = 0.178) revealed for stride length. The main effects of slope were not significant. Stride length was longer during uphill UL and FL walking compared to level walking at the same carrying modes and lower during FL walking compared to UL and BL walking regardless of the slope.

The main effects of the loading condition (F = 49,035, *p* ≤ 0.001, partial η^2^ = 0.628) and a significant slope-by-loading condition interaction (F = 6628, *p* ≤ 0.001, η^2^ = 0.186) were significant for cadence. The main effects of slope were not significant. Cadence was lower during walking at slopes 5, 10, and 15% compared to level (0%) walking for the UL and FL carrying mode and greater during level and uphill FL walking compared to UL and BL walking (see pairwise comparisons in Table 2).

Step and stride length were significant longer and cadence significant lower during BL walking compared to UL walking, but only on 0% slope with differences with other slopes being not significant.

### 3.4. Heart Rate and Perceived Exertion

Significant main effects of slope (F = 152,754, *p* ≤ 0.001, partial η^2^ = 0.840) and loading condition (F = 37,240, *p* ≤ 0.001, partial η^2^ = 0.562) were revealed for HR. Heart rate was gradually increased with increased slope in all loading conditions (*p* ≤ 0.001) and it was higher during both BL and FL walking compared to the UL walking (see pairwise comparisons in Table 3).

The main effects of slope (F = 69,699, *p* ≤ 0.001, partial η^2^ = 0.706) and loading condition (F = 40,433, *p* ≤ 0.001, partial η^2^ = 0.582) were significant for PE. Post hoc analysis showed that PE was gradually increased with increased slope in all loading conditions (*p* ≤ 0.001). Higher PE was also obtained during both uphill BL and FL walking compared to UL walking (see pairwise comparisons in Table 3).

## 4. Discussion

### 4.1. Effects on EMG Activity

The results of the study showed that the EMG activity of the CES, TES, and LES increased as the slope of the treadmill’s surface increased, with uphill FL walking eliciting higher EMG activity for all parts of erector spinae compared to the UL and BL walking. Uphill BL walking, on the other hand, showed that while it caused an increase in the activity of the CES, it decreased the activity of the LES without affecting the activity of the TES compared to the UL walking condition. Furthermore, trunk flexor activation by means of the EMG activity of RAB was equally high during uphill BL and FL walking compared to uphill UL walking but increased progressively with the slope of the treadmill’s surface only during the FL walking condition.

Postural responses in the cervical spine during level and uphill unloaded and loaded walking have been investigated by many researchers in terms of changes in the position of the head and neck [44,45,46,47] with information on the activation of the CES during walking, particularly when carrying a load uphill still lacking. Cromwell [46] showed that the trunk is flexed, and the neck is practically extended during level walking causing a flexed head position to maintain the orientation of the head within the base of support. The same authors reported that the head is flexed to a greater extent during uphill walking on a ramp pitched at 8.5° than during horizontal walking, shifting its center of mass further forward, but was lagged relative to the forward motion of the neck, compensating for the fact that neck led the forward tilt of the trunk. Even though postural changes were not investigated in the present study it can be assumed that the progressive increase in the EMG activity of the CES during uphill unloaded walking, is the subsequent manifestation of the compensatory response of these muscles to maintain the head vertically aligned over the trunk [46].

Our findings also revealed that the activity of CES was affected by the load carrying mode. Uphill FL walking elicited higher activity of CES compared to uphill BL walking with both loaded walking conditions eliciting higher activity compared to uphill UL walking. Differences between loading modes in CES activity may be due to further changes in the posture of the head and neck that occur in the sagittal plane to compensate for the elicited forward tilt of the trunk [48,49,50]. Chansirinukor et al. [45] have noted an increase in forward head posture when a backpack of 15% body weight (BW) was carried by students during a continuous 5 min level walk. Other authors confirmed these findings [44,47] adding that level FL walking at 0.75 stride s^−^^1^ resulted in a more upright posture of the head [44] possibly caused by a reduced forward trunk tilt. Although these studies investigated the compensatory responses of the head and neck during level loaded walking, it is likely that similar responses may occur during uphill loaded walking [49] necessitating higher CES EMG activity to maintain the head in a more upright posture than to hold it against gravity as occur during uphill FL and BL walking, respectively.

Previous studies showing increased and decreased LES activity during level FL [17,22] and level BL walking [17,18,20], respectively, compared to level UL walking agree with our findings. The increase in TES and LES muscle activity found in the present study with uphill UL and BL walking was also consistent with the increase in superficial and deep trunk extensors muscle activity found by other authors investigating similar walking conditions [23,24,25,26]. These muscles, namely the thoracic and lumbar erector spinae as well as multifidus, play an important role in resisting the bending forces and the associated shear and compression forces created in the lumbar spine during forward trunk tilt. It is thought that this occurs either by storing elastic energy within the muscle as the muscles’ length increases with flexion of the lumbar spine [51] or by optimizing the muscle’s length–tension relationship, which increases linearly up to 45° of trunk flexion [52].

The progressive increase in TES and LES muscle activity during uphill loaded walking could be attributed to the increased demands placed on these muscles for compensating for the bending moment resulting from the gradual increase of the treadmill’s surface slope and the load being carried either in front or the back of the trunk. Uphill UL walking linearly increases forward trunk tilt [3,5,53] reaching from approximately 10° during the stance phase of walking on 0° (level), to slightly over 20° during the same phase of walking on 15° inclination [3]. This postural response may justify an increase in the activity of the anti-gravity trunk muscles, as they are required to produce the necessary force to hold the head–arm–trunk (HAT) against gravity during uphill UL walking [23,24,25]. The suspension of the load in front of the trunk was expected to further shift the center of mass of the HAT forward due to the volume of the added load in the form of the books. This projection could range from 10.8 to 19.9 cm, considering the weight-to-thickness ratio of the books (i.e., 1/1.5) used to achieve the appropriate carried load (15% BW) for a person with body mass as low as 48 kg and as high as 88.5 kg. Therefore, more effort would have to be exerted by TES and LES to compensate for the bending moment induced by the added load, which ultimately manifested with the increase in EMG muscle activity. In contrast, the EMG activity produced by the LES during uphill BL walking is lower compared to uphill UL walking. The reduced activity of the lumbar erector spinae observed during uphill BL walking is probably due to the need to compensate for a lower bending moment resulted by the application of a load to the posterior surface of the trunk. This load creates an extension moment that potentially reduces the bending moment generated by the HAT [18].

Previous studies have shown that the EMG activity of the RAB is relatively low (<5% of MIVC) throughout the gait cycle of level walking, contributing minimally to the stabilization of the trunk [54]. Other investigators showed that uphill walking on a treadmill at a 7° slope [55] with the preferred speed (the highest speed at which the participants could walk naturally) or ascending a ramp with a slope of 15° [23] did not alter RAB activation compared to level walking. These results agree with the findings of the present study which showed that RAB activity remained unaffected by the slope during uphill UL walking. Our findings are also similar to those of other studies which have shown that level BL [20] and uphill BL walking [28] elicited higher EMG activity of the RAB than the unloaded and the level walking condition, respectively. The activity of the RAB during FL walking was also greater than that elicited during UL walking and increased progressively with increased slope. This is an indication of an increasing demand for dynamic stabilization during this loaded condition. However, although RAB activity increased on average by approximately 33.1% and 49.9% during BL and FL walking, respectively, relative to the UL walking conditions, it did not exceed 7% of MIVC (at 15% slope FL walking), indicating that they contribute little to trunk stabilization during uphill loaded walking. Clinically, insufficient co-contraction between the LES and RAB during BL walking indicates that the trunk lacks dynamic stability, and the load is carried passively, a fact which may prove detrimental to the passively stabilizing structures of the spine. The imbalance between the LES and the RAB during FL walking, on the other hand, even though the EMG activity of the latter gradually increases with the slope of the surface, leaves LES to bear the exerted load and thus be exposed to fatigue, deterioration, and pain.

### 4.2. Effects on Trunk Sway

The results of the present study showed that the projected 95% ellipse area and AP trunk displacement gradually increased as the slope of the treadmill’s surface increased during uphill UL, BL, and FL walking. The projected ML trunk displacement however, remained unaffected. Clinically these findings suggest that walking uphill particularly carrying a front load can compromise body stability, increasing the risk of injury (e.g., fall). The observed changes could partly be explained by the metabolic and sensorial changes induced by walking. Metabolically, postural control may be affected by the increase in heart and respiratory rate that follows a general muscular type of exercise such as walking, particularly when it is performed vigorously (e.g., uphill). Walking uphill, as has been proposed for running, may also affect interoceptive and exteroceptive information and/or its integration, disrupting postural control. Although visual and proprioceptive inputs initially compensate for this disturbance, body dehydration and metabolic processes occurring in the fatiguing muscles further impair postural control, which subsequently can be controlled by compensatory motor strategies and increased cognitive contribution. As most of these factors were not measured in the present study, they will not be discussed further. Readers are encouraged to study the review of Paillard [56], in which the contribution of these parameters in body sway disturbances is presented in detail.

The projected AP trunk displacement may have been further affected, albeit not significantly, by the load carrying mode, being higher during uphill FL walking compared to uphill UL and BL walking. These differences could be partially explained by changes in spatiotemporal gait parameters such as cadence, which was higher during uphill FL walking compared to uphill UL and BL walking. The higher cadence, possibly compensating for the shorter step/stride, may have caused more trunk sway in the subject’s attempt to maintain the body’s center of mass within the base of support.

Published data on body stability under loaded walking conditions support to some extent our findings, although most of them have been collected during level walking. In agreement with our findings, a previous study [30] showed that carrying a 30 lb. load in the front or back of the trunk during over ground walking at a self-selected pace resulted in a significant increase in AP center of mass displacement compared to unloaded walking. Simpkins et al., however, showed recently that anterior load carriage did not affect the position and the velocity of body-load system’s center of mass during treadmill level walking at 1.2 m/s [29]. Regarding ML trunk displacement some authors have reported significant effects as a result of level FL walking [30] and others have shown no-significant changes during uphill BL walking, a finding that agrees with the findings of the current study [28]. The differences between the present and other studies may be due to the fact that in our study participants were only selected if they demonstrated musculoskeletal symmetry bilaterally as well as due to the experimental conditions used for testing. For example, postural responses may differ between studies, as in our study the inertial sensor was placed on the upper torso, as opposed to what other researchers do by placing it approximately at the level of the body’s center of mass.

### 4.3. Effects on Spatiotemporal Gait Parameters

Uphill UL walking resulted in a slight but significant increase in step/stride length (<1.0 cm on average) and a decrease in cadence (<1.0 strides min^−^^1^) compared to level UL walking, with all spatiotemporal parameters remained unaffected during uphill walking at slopes between 5 and 15%. Comparing our findings with those of other studies, it was found that there was agreement on the decrease in cadence [57,58,59], while the evidence on step/stride length varied, with some researchers reporting that it remained unaffected between 0° and 10° of uphill walking [60] and others that it increased [57] or decreased with increasing slope [58,59].

Step/stride length was increased, and cadence was decreased during FL walking at 5% compared to 0%, remaining thereafter unchanged. Furthermore, step/stride length was lower and cadence greater during FL walking compared to both UL and BL walking regardless of the slope. Shorter stride length and higher cadence have also been obtained by other authors when carrying a front load of 10–20% of body weight compared to an unloaded condition, but only during level walking [30,61]. Step/stride length and cadence were not affected by the slope during BL walking which agrees with the findings reported in a previous study [60]. Liu et al. [60] in this study showed that step length remained unaffected during treadmill walking at 3°, 5°, and 10° with 4 km h^−^^1^ carrying backpack loads of 10, 20, and 30 kg compared to level (0°) walking without backpack loads. Spatiotemporal gait parameters were also not significant different between the BL and UL walking condition in all but the 0% slope, suggesting that as far as these parameters are concerned, gait biomechanics is not affected by the BL carriage.

Differences between uphill FL and BL walking regarding step/stride length and cadence may be explained by postural changes of the trunk (e.g., forward trunk tilt). Trunk TES and LES EMG activity was increased during uphill FL walking, suggesting that there was an increased need for maintaining the trunk in a less flexed posture compared to uphill BL walking [49]. Consequently, participants may have been forced to shorten the step/stride length increasing subsequently cadence, to keep up the preselected and stable speed of the treadmill and thus maintain the center of the body-load mass on the base of support. Conversely, by achieving greater trunk flexion during BL walking [48,49,50], participants may have adopted a longer step/stride length, which led them to reduce the cadence preventing them from falling.

### 4.4. Practical Implications

Uphill walking with or without carrying a load is a common activity performed by many individuals. The slopes of the walking surface as well as the magnitude and modes of the load carried in the present study were selected among the most frequently encountered both in daily life and recreation as well as part of work tasks in several occupations. The speed at which walking was performed, on the other hand, was selected based on the average comfortable speed used by males and females for level walking and was held constant during level and uphill BL and FL walking. This decision was based on the authors’ intention to minimize the effects of other potential sources of influence, rather than the independent variables themselves (e.g., slope and loading mode) on muscle activity, trunk sway, and gait parameters. Furthermore, participants in this study were physical education college students and therefore expected to have a satisfactory level of physical fitness to face the demands of the study protocol [62]. This was also supported by the fact that the participants indicated an average of 13 points on the Borg scale, which means that they perceived the effort as “somewhat hard”. Nevertheless, the variability in perceived exertion demonstrated by the study sample (SD: ±2.9 points) suggest that some of them experienced discomfort and/or fatigue due to continuous walking on a level and sloped surface at the same speed.

However, it is not uncommon for citizens, recreational athletes, and professionals to perform real-life activities at a constant high speed such as that applied in the present study protocol, even though it is not the speed typically chosen to perform such activities. Gupta et al. [63] showed that male and female pedestrians 16–25 years of age (young adults) in Dharamshala, a famous touristic city in India with an average elevation of 1457 m, increased their walking speed when the slope of the road was increased from 3 to 9%. Although the average speed recorded in this age group while walking at 9% slope was approximately 4 km h^−^^1^ (66.2 m min^−^^1^) some individuals reached 5.1 km h^−^^1^ (84.9 m min^−^^1^) as indicated by the highest walking speed recorded in this slope. Interestingly, even though the average speeds achieved by elder adults (26 to 50 years) while walking at the same slopes was lower compared to young adults (3.4–3.7 km h^−^^1^ or 57.4–61.8 m·min^−^^1^) some individuals reached speeds of 5.8 km h^−^^1^ (96.0 m min^−^^1^) while walking at 9% slope [63]. Moreover, the walking speed of pedestrians regardless of gender and age up to >50 years who carried baggage at the same slope could reach 4.9 km h^−^^1^ (80.9 m min^−^^1^) speed. The speeds achieved by citizens of economically developed countries may be even higher while walking under these conditions, as their walking speeds tend to be higher than the walking speeds of citizens of less economically developed countries such as India [64]. Backpackers also tend to walk at a speed of 3.2 to 4.8 km h^−^^1^ (2–3 mph) on trails with various slopes, with some of them (thru-hikers) adopting a more powerful way of hiking (4.8–6.4 km h^−^^1^) (https://trailandsummit.com (accessed on 21 October 2022), while trail runners switch to power hiking to reduce their energy expenditure when they face steep slopes, although an individual’s biomechanics may require trail running-to-power hiking switch as a different slope. Walking speeds of 5.5 km h^−^^1^ and as high as 7.5 km h^−^^1^ are also required for unloaded and loaded speed marching, respectively [65,66], i.e., common military exercises carried out by marching relatively fast over a distance and at various slopes, without and with carrying a load. The above findings showed that walking fast at a slope may not be as uncommon as expected, particularly by persons who walk aiming at a target. This is supported by previous findings showing that “attentional narrowing”, whereby spatial perception is affected by the visual focus of attention on a specific target, brings one closer to the “finish line” of an event. Attentional narrowing eventually increases walking speed and the subjective perception of performing physical tasks with ease compared to people looking at the environment as they naturally would [67]. People carrying a front load, such as a baby in a baby carrier or an object, may also maintain a steady walking speed while walking on level and sloped surfaces, particularly if they also adopt narrow attention. This is not unexpected considering the city and work environments in which a person moves combined with the stress and time constraints under which many daily activities and professional tasks are carried out.

### 4.5. Limitations

Two factors that could affect the results limiting their generalizability are perceived fatigue and heart rate that increased during walking with a progressively increasing slope, showing higher values during uphill BL and FL walking than during uphill UL walking. Both factors may partially affect test results, as they were expected to be influenced by test conditions such as test duration and the preselected speed. Although erector spinae have a large percentage of type I fiber [68] and high vascularization [69], making it highly efficient in lumbar activities that require high levels of muscular endurance, their activity could have been affected as they had to sustain the suspended load for at least 10 min in each loading condition. Similar responses have been reported in a previous study where Indian soldiers had to walk on different slopes for the same amount of time [26]. Furthermore, heart rate may have also influenced trunk sway data as it exceeded 60% of maximum heart rate, a limit that when exceeded can affect postural control [56,70]. Since these factors are intrinsically related to the effort required to perform such activities, they should be considered when attempting to explain the elicited responses.

The results of our study could also be different if the participants were professionals or workers with experience in carrying heavy loads, school-aged or elderly people, or individuals with health problems [71,72]. Moreover, our results are limited to the testing procedure characteristics, such as the slope and the type of the surface used for walking, the weight of the suspended load, and the carrying mode [31,32].

## 5. Conclusions

The results of this study showed that uphill unloaded and loaded walking is a demanding activity that requires a gradual increase in the activity of CES, TES, and LES. Overall, erectors spinae’s EMG activity is greater when carrying a front pack than a backpack, the long-term performance of which can cause muscle fatigue and dysfunction. The increased AP trunk sway, possibly as a result of increased cadence coupled with reduced step/stride length that individuals adopt during uphill FL walking, may also compromise body stability, leading to fall-related injuries. Based on the current findings, uphill walking carrying a load on the back of the trunk is the least demanding activity in terms of trunk muscle activation and body stabilization. This information may contribute to optimizing performance and preventing possible injuries that occur in individuals involved in performing such activities.

## Figures and Tables

**Figure 1 sensors-23-00609-f001:**
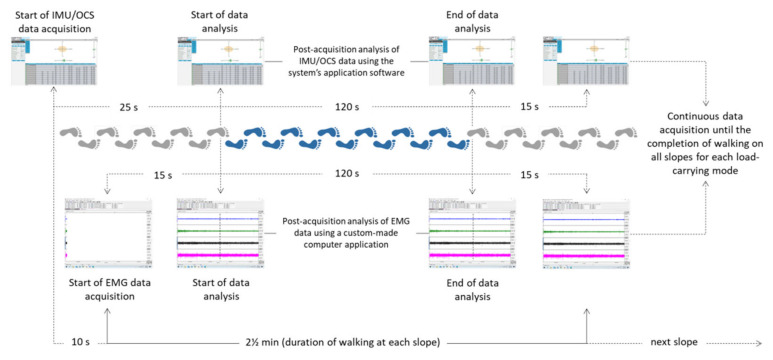
Graphical representation of the setup used to simultaneously acquire and record IMU/OSC and EMG data. Data acquisition, once started, continued without interruption while walking on all four slopes at all loading conditions. Post-acquisition data analysis was performed based on the number of steps performed in the intermediate 120 s of total walking duration in each slope.

**Figure 2 sensors-23-00609-f002:**
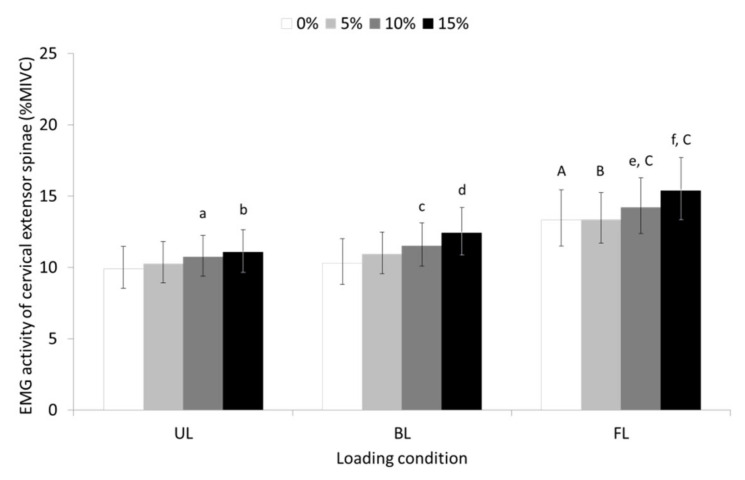
Geometric means and 95% confidence intervals (error bars) for the EMG activity of cervical erector spinae during unloaded (UL), back-loaded (BL), and front-loaded (FL) walking at 0 (level), 5, 10, and 15% uphill slope. ^a^ significant different (SD) compared to 0% slope (*p* ≤ 0.05); ^b^ SD compared to 0 (*p* ≤ 0.001) and 5% (*p* ≤ 0.05); ^c^ SD compared to 0 (*p* ≤ 0.05) and 5% (*p* ≤ 0.01) slope; ^d^ SD compared to 0 and 5% slope (*p* ≤ 0.001) as well as 10% (*p* ≤ 0.01); ^e^ SD compared to 0% (*p* ≤ 0.01); ^f^ SD compared to 0% (*p* ≤ 0.001) as well as 5 and 10% slope (*p* ≤ 0.01); ^A^ SD compared to UL and BL condition (*p* ≤ 0.001); ^B^ SD compared to UL (*p* ≤ 0.001) and BL condition (*p* ≤ 0.05); ^C^ SD compared to UL (*p* ≤ 0.001) and BL condition (*p* ≤ 0.01).

**Figure 3 sensors-23-00609-f003:**
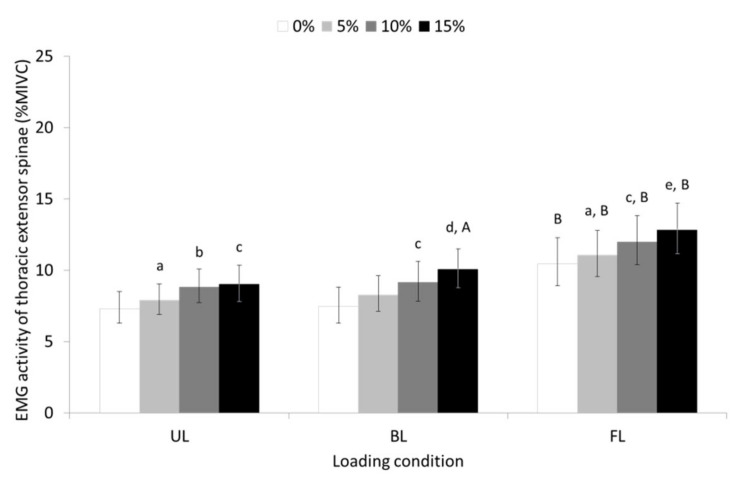
Geometric means and 95% confidence intervals (error bars) for the EMG activity of the thoracic erector spinae during unloaded (UL), back-loaded (BL), and front-loaded (FL) walking at 0 (level), 5, 10, and 15% uphill slope. ^a^ significant different (SD) compared to 0% slope (*p* ≤ 0.01); ^b^ SD compared to 0% (*p* ≤ 0.001) and 5% slope (*p* ≤ 0.05) ^c^ SD compared to 0 and 5% slope (*p* ≤ 0.001); ^d^ SD compared to 0, and 5% slope (*p* ≤ 0.001) as well as 10% slope (*p* ≤ 0.05); ^e^ SD compared to 0, and 5% slope (*p* ≤ 0.001) as well as 10% slope (*p* ≤ 0.01); ^A^ SD compared to UL condition (*p* ≤ 0.05); ^B^ SD compared to UL and BL condition (*p* ≤ 0.001).

**Figure 4 sensors-23-00609-f004:**
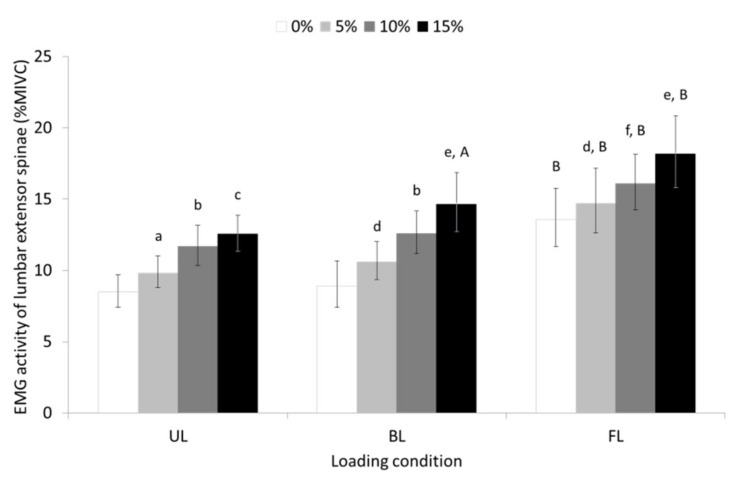
Geometric means and 95% confidence intervals (error bars) for the EMG activity of the lumbar erector spinae during unloaded (UL), back-loaded (BL), and front-loaded (FL) walking at 0 (level), 5, 10, and 15% uphill slope. ^a^ significant different (SD) compared to 0% slope (*p* ≤ 0.001); ^b^ SD compared to 0 and 5% slope (*p* ≤ 0.001); ^c^ SD compared to 0 and 5% (*p* ≤ 0.001) as well as 10% slope (*p* ≤ 0.05); ^d^ SD compared to 0% slope (*p* ≤ 0.05); ^e^ SD compared to 0, 5 and 10% slope (*p* ≤ 0.001); ^f^ SD compared to 0% (*p* ≤ 0.001) and 5% slope (*p* ≤ 0.05); ^A^ SD compared to UL condition (*p* ≤ 0.05); ^B^ SD compared to UL (*p* ≤ 0.001) and BL condition (*p* ≤ 0.01 in all but 15% slope where *p* ≤ 0.05).

**Figure 5 sensors-23-00609-f005:**
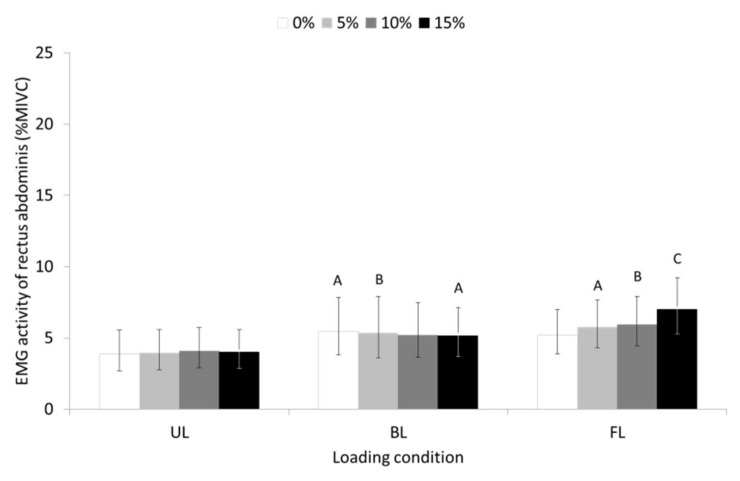
Geometric means and 95% confidence intervals (error bars) for the EMG activity of rectus abdominis during unloaded (UL), back-loaded (BL), and front-loaded (FL) walking at 0 (level), 5, 10, and 15% uphill slope. ^A^ SD compared to UL condition (*p* ≤ 0.05); ^B^ SD compared to UL condition (*p* ≤ 0.01); ^C^ SD compared to UL condition (*p* ≤ 0.001).

**Figure 6 sensors-23-00609-f006:**
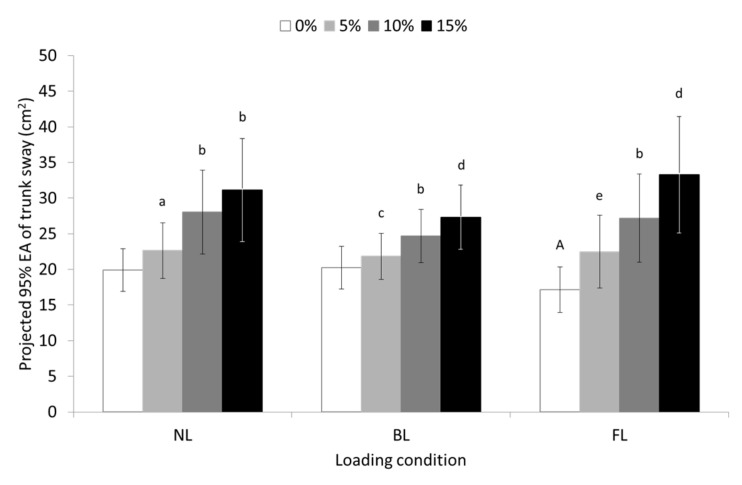
Means and 95% confidence intervals (error bars) for the projected trunk sway’s 95% ellipse area during unloaded (UL), back-loaded (BL), and front-loaded (FL) walking at 0 (level), 5, 10, and 15% uphill slope. ^a^ significant different (SD) compared to 0% slope (*p* ≤ 0.01); ^b^ SD compared to 0 and 5% slope (*p* ≤ 0.001); ^c^ SD compared to 0% slope (*p* ≤ 0.05); ^d^ SD compared to 0 (*p* ≤ 0.001) and 5% slope (*p* ≤ 0.01); ^e^ SD compared to 0% (*p* ≤ 0.001); ^A^ SD compared to UL (*p* ≤ 0.05) and marginally SD compared to BL condition (*p* = 0.054).

**Figure 7 sensors-23-00609-f007:**
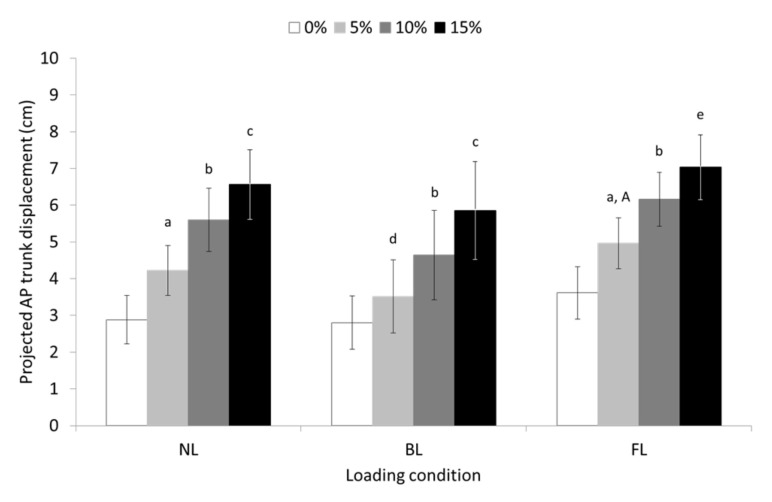
Means and 95% confidence intervals (error bars) for the projected anteroposterior (AP) trunk displacement, during unloaded (UL), back-loaded (BL), and front-loaded (FL) walking at 0 (level), 5, 10, and 15% uphill slope. ^a^ significant different (SD) compared to 0% slope (*p* ≤ 0.001); ^b^ SD compared to 0 and 5% slope (*p* ≤ 0.001); ^c^ SD compared to 0, 5 and 10% slope (*p* ≤ 0.001); ^d^ SD compared to 0%, (*p* ≤ 0.05); ^e^ SD compared to 0, and 5 % slope (*p* ≤ 0.001) as well as 10% slope (*p* ≤ 0.01); ^A^ SD compared to BL condition (*p* ≤ 0.05).

**Figure 8 sensors-23-00609-f008:**
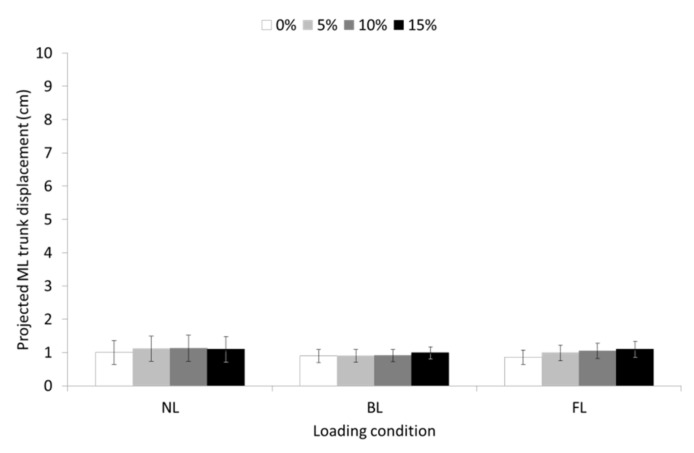
Means and 95% confidence intervals (error bars) for the projected mediolateral (ML) trunk displacement during unloaded (UL), back-loaded (BL), and front-loaded (FL) walking at 0 (level), 5, 10, and 15% uphill slope.

**Table 1 sensors-23-00609-t001:** Means ± standard deviations of the anthropometric and musculoskeletal characteristics of the study sample.

Anthropometric Characteristics	Males	Females	Total
Age (years)	25.6 ± 3.3	25.5 ± 3.1	25.6 ± 3.3
Height (m)	1.8 ± 0.1	1.6 ± 0.1	1.7 ± 0.1
Body mass (kg)	75.5 ± 7.6	60.2 ± 8.5	67.9 ± 11.1
Body mass index (kg m^−2^)	23.8 ± 2.1	22.2 ± 3.2	23.0 ± 2.8
**Musculoskeletal characteristics**			
Thoracic curve (°)	33.0 ± 6.2	32.7 ± 9.1	32.8 ± 7.6
Trunk rotation (°)	1.9 ± 0.8	2.6 ± 1.0	2.3 ± 1.0
Shoulder protraction asymmetry (cm)	0.8 ± 0.7	0.8 ± 0.8	0.8 ± 0.7
Leg length discrepancy (cm)	0.1 ± 0.2	0.1 ± 0.2	0.1 ± 0.2
Pelvic tilt (°)	0.9 ± 0.6	0.9 ± 0.8	0.9 ± 0.9
Pelvic rotation (°)	2.7 ± 1.9	2.5 ± 1.4	2.6 ± 2.6

**Table 2 sensors-23-00609-t002:** Means ± standard deviations of spatiotemporal gait parameters during unloaded (UL), back-loaded (BL), and front-loaded (FL) gait at 0% (level), 5%, 10%, and 15% uphill slope.

Spatiotemporal Gait Parameters	Slope		Loading Condition	
		UL	BL	FL
Step length (cm)	0%	78.6 ± 4.1 ^a^	79.3 ± 4.3 ^A^	76.6 ± 4.3 ^b, B^
	5%	79.3 ± 4.2	79.3 ± 4.3	77.6 ± 4.4 ^B^
	10%	79.4 ± 4.3	79.1 ± 4.2	77.9 ± 4.4 ^B^
	15%	79.2 ± 4.2	78.9 ± 4.5	77.8 ± 4.4 ^B^
Stride length (cm)	0%	157.2 ± 8.2 ^a^	158.5 ± 8.7 ^A^	153.2 ± 8.5 ^b, B^
	5%	158.5 ± 8.4	158.6 ± 8.6	155.3 ± 8.7 ^B^
	10%	158.8 ± 8.6	158.3 ± 8.4	155.9 ± 8.8 ^B^
	15%	158.3 ± 8.4	157.8 ± 8.9	155.6 ± 8.9 ^B^
Cadence (step/min)	0%	106.2 ± 5.7 ^a^	105.3 ± 6.0 ^A^	109.0 ± 6.2 ^b, B^
	5%	105.3 ± 5.7	105.3 ± 5.9	107.6 ± 6.3 ^B^
	10%	105.1 ± 5.8	105.5 ± 5.8	107.2 ± 6.3 ^B^
	15%	105.4 ± 5.8	105.9 ± 6.2	107.4 ± 6.4 ^B^

^a^ Significant different (SD) compared to 5% slope (*p* ≤ 0.01); ^b^ SD compared to 5 and 10% (*p* ≤ 0.001) as well as 15% slope (*p* ≤ 0.05); ^A^ SD compared to UL (*p* ≤ 0.05); ^B^ SD compared to UL and BL (*p* ≤ 0.001).

**Table 3 sensors-23-00609-t003:** Means ± standard deviations of heart rate and perceived exertion during unloaded (UL), back-loaded (BL), and front-loaded (FL) walking at 0 (level), 5, 10, and 15% uphill slope.

Fatigue-Related Parameters	Slope	Loading Condition		
		UL	BL	FL
HR (bpm)	0%	98.8 ± 15.6 ^a^	109.8 ± 14.9 ^a, A^	110.8 ± 15.9 ^a, A^
	5%	109.1 ± 14.9 ^b^	118.3 ± 15.7 ^b, A^	119.4 ± 15.3 ^b, A^
	10%	117.1 ± 14.4 ^c^	125.9 ± 17.1 ^c, A^	127.7 ± 17.0 ^c, A^
	15%	126.4 ± 15.3	135.0 ± 19.4 ^A^	136.1 ± 17.0 ^A^
PE (points)	0%	7.30 ± 1.1 ^a^	9.4 ± 2.4 ^a, A^	9.4 ± 2.3 ^a, A^
	5%	8.0 ± 1.2 ^b^	10.4 ± 2.2 ^b, A^	10.5 ± 2.4 ^b, A^
	10%	9.2 ± 1.8 ^c^	11.6 ± 2.5 ^c, A^	11.9 ± 2.6 ^c, A^
	15%	10.4 ± 2.2	13.0 ± 2.9 ^A^	13.1 ± 2.9 ^A^

HR: Heart rate in beats per minute (bpm); PE: Perceived exertion measured by the numbered 6–20 Borg’s 15-point rating scale; ^a^ Significant different (SD) compared to 5, 10 and 15% (*p* ≤ 0.001); ^b^ SD compared to 10 and 15% (*p* ≤ 0.001); ^c^ SD compared to 15% (*p* ≤ 0.001); ^A^ SD compared to UL condition (*p* ≤ 0.001).

## Data Availability

The data presented in this study are available upon request from the corresponding author.

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
