# Peer review of "Effects of Load Carriage on Postural Control and Spatiotemporal Gait Parameters during Level and Uphill Walking"

_sensors, 2023, doi:10.3390/s23020609_

Round 1
Reviewer 1 Report
Manuscript# sensors-1867682
Title: Effects of load carriage on postural control and spatiotemporal parameters during level and uphill walking
General Comments
The goal of this study was to investigate the effect of uphill front-loaded or back-loaded walking on postural control and spatiotemporal parameters of gait. The authors reported that EMG uphill front-loaded walking increased EMG activity of trunk extensors and flexors and body sway, being greater compared to uphill unloaded or back-loaded walking. Step and stride length was greater and cadence was lower during uphill front-loaded walking than level walking. Step and stride length were lower and cadence greater for uphill front-loaded walking than for unloaded or back-loading walking. The authors concluded that postural control and spatiotemporal gait parameters are compromised during level and uphill walking when carrying a front load presumably increasing the risk for falling related injury. While this this manuscript is generally well written, there are several major aspects requiring revision:
Comments
1. Abstract: Please include the objectives of this study.
2. Line 74: It is unclear how the variables related to body sway during standing posture can be transferred to walking. Please clarify and provide adequate references showing that this is adequate.
3. Lines 92-93: The relevance of the data outlined here is a far fetch. Please revise.
4. Methods: Several aspects in the methods section require additional information. Please keep in mind that sufficient information (or references) must be provided that would allow the reader to replicate the study.
a. The walking speed of 5 km/h uphill with load carriage is very fast. In fact, according to Table 3, some participants reached an exertion of 15 on a 15 point Borg scale. The relevance of data at this exertion state is questionable as most people would not choose to perform at this exertion level. This aspect should be included in the discussion section. The total duration of the experiment was 40 minutes. Although the authors randomized the order of conditions, there is a substantial fatigue effect that may explain the large variability in data among participants.
b. Please describe how an interference of the IMU and the EMG electrodes with the front/backpack was avoided.
c. How were left and right side muscles included in the analysis?
d. The data processing must be described in greater detail. Was the EMG and the IMU data synchronized? Was the data normalized to gait cycle? Please show the EMG traces and projected body sway data. The authors must be able to understand the calculation of output parameters.
e. Was the IMU validated for the parameters of interest? Please provide data or appropriate references.
f. The incline was increased every 2.5 minutes. During which period were parameters calculated?
g. Statistical analysis: It is not sufficient to report a theoretical sample size calculation. Based on what information was the effect size assumed?
5. Line 239: loaded walking increased – there was no load increase but three different loading conditions.
Specific comments
6. Line 19: This sentence lacks the comparator. Please revise.
7. Line 28: Please define CNS.
8. Line 32 and 33: Please remove ”the”.
9. Line 41: Please revise this sentence to clarify that this is hypothetical or otherwise provide references for this statement.
10. Line 46: “choose to be actively transported” – revise.
11. Line 48: Trailism = an alternative interpretation of mind-body dualism of Descartes – please revise accordingly.
12. Line 49: unless the leave on steep slope – unclear.
13. Line 56: may justify such injures – revise.
14. Line 59: possibly completely interrupt blood flow – Did this reference really show this?
15. Line 80: Based on systematically reviewed data – please revise.
16. Lines 86-89: This section is unclear. Please revise.
17. Table 1 is not cited in the text.
18. Table 1: Do not appreviate years. Body weight is measured in N, body mass in kg.
19. Line 118: anterior superior iliac spine.
20. Table 2: Please ensure that . are used as decimal separators.
21. Table 2: Please report data with an adequate level of accuracy (currently 1/10 of a mm).
22. Figure 7: Please scale the figure appropriately.
23. Line 423-426: This is a hypothetical statement. The authors did not measure moments. Please provide adequate references. Please focus on the reported data. Same for Line 433-435
24. Please copy-edit the manuscript with a focus on commata and articles.
Author Response
Dear Editor-in-Chief, Dear reviewer
I would like to thank you for the opportunity to resubmit the article with the corrections you requested. An attempt has been made to respond to all comments as adequately as possible. However, we are willing to respond to further comments if any of them have not been understood or adequately answered.
Please find attached a cover letter listing all responses to your comments.
Sincerely

Reviewer 2 Report
Dear authors,
Thank you for submitting your manuscript for review. Please find below some commentaries and suggestions.
Abstract: the gait velocity is presented as 5 km h-1. I believe a dot between km and h is missing. The same happens in the rest of the manuscript.
Abstract: in the same same line it reads “with suspended a frontpack”. Would it not be “with a suspended frontpack”?
Line 28: CNS abbreviation is used, but its meaning was not yet introduced.
Line 36: “day backs” or “day packs”?
Line 124: ASIS abbreviation has not been introduced.
Line 140: while the authors have provided information regarding the backpack used, it would be interesting to know a little more. Since the backpack was filled with books to achieve the necessary percentage of body weight, it is reasonable to assume that the number and weight of books varied. This may have resulted in more or less books distributed along the backpack “depth” (direction normal/perpendicular to the back/chest). This could have influenced the torque (force * distance) produced by the pack, and therefore the muscular response. Is there any model/manufacturer information?
Line 166: the authors recorded the EMG activity with a sampling frequency of 1000 Hz, and filtered the signal with a 10-1000 Hz FIR filter. However, and considering the Nyquist Theorem, the maximum sampled frequency (considering the sampling frequency) would have been 500 Hz. Most softwares, including the Biopac Acknowledg, does not allow filtering above the Nyquist Frequency (half the sampling frequency). How did the authors perform this filtering? Was a digital filter been used, or an analog (physical) filter used? In case of an analog filter, which is its characteristics?
Line 167: what was the author’s rationale for using a 50 Hz notch filter? The 50 Hz frequency unfortunately lies on the EMG band of interest (roughly 20-450 Hz). However, these filters can be highly destructive of signal information, especially when the notch band is too wide or the filter order too low. An explanation on why this filter was used, and more information on the filter’s characteristics would be valuable to understand its implications.
Line 172-173: I believe this was done to obtain a RMS-envelope of the signal. Was the signal previously full-wave rectified, or only the positive portion of the EMG signal was enveloped? I understand the RMS calculation by definition should result in a full-wave rectification of the signal, but If the authors used Biopac Acknowledg software, I believe that full-wave rectification is optional in the dialog box.
Figure 1: the mean and confidence intervals do not seem to be higher than 15% in any condition. Is there any reason to have a yy axis scaled to 25%? Could it be reduced to 20% or even 15% so that the differences between conditions could be better appreciated? Perhaps to 20% since it seems to be the minimum common value for all Figures of EMG.
Line 292: until now, the authors have been referring to body sway along the manuscript, although the inertial sensor being placed between the scapulas, which is not a good approximation to the body’s centre of mass, and expected to provide higher AP and ML displacements due to the increased distance from the pelvis. Here, the authors refer for the first time to “trunk sway”. This seems a much more accurate description of what was indeed performed. I would suggest the authors to review the manuscript and refer to the sway measurement as trunk sway, and not body sway. If the authors believe that this approach is still reliable as a body sway measurement, I would appreciate some justification or supporting reference.
Figure 7: I believe the authors have decided to use the same yy axis scale in all figures reporting similar parameters. However, in this case, the -10 to +10 scale is notoriously excessive, and the reader cannot grasp the differences/similarities between conditions. This is particularly important considering there is no other source of information for these results (i.e. a table).
Table 2: there is an inconsistent use of dot (.) and comma (,) as a decimal separator.
Line 514: “step/stride length” instead of “stride/stride length”.
Line 554: “step/stride length” instead of “stride/stride length”.
Author Response

(The authors gave the same response as above.)

Round 2
Reviewer 1 Report
Manuscript# sensors-1867682
Title: Effects of load carriage on postural control and spatiotemporal parameters during level and uphill walking
General Comments
The goal of this study was to investigate the effect of uphill front-loaded or back-loaded walking on postural control and spatiotemporal parameters of gait. The authors reported that EMG uphill front-loaded walking increased EMG activity of trunk extensors and flexors and body sway, being greater compared to uphill unloaded or back-loaded walking. Step and stride length was greater and cadence was lower during uphill front-loaded walking than level walking. Step and stride length were lower and cadence greater for uphill front-loaded walking than for unloaded or back-loading walking. The authors concluded that postural control and spatiotemporal gait parameters are compromised during level and uphill walking when carrying a front load presumably increasing the risk for falling related injury.
While the authors have addressed most comments, several major methodological concerns remain:
Comments
1. As stated in the first review, the walking speed of 5 km/h uphill with load carriage is very fast. Although the authors added the aspect of fatigue in their limitations, the relevance of data at this exertion state is still questionable as most people would not choose to perform at this exertion level. This aspect must be included in the discussion section. The total duration of the experiment was 40 minutes. Although the authors randomized the order of conditions, there is a substantial fatigue effect that may explain the large variability in data among participants.
2. The information regarding the avoidance of interference of the IMU and the EMG electrodes with the front/backpack is not sufficient. Considering the muscles assessed, the backpack must have been placed on top of the sensors. Presumably, this placement will lead to artifacts in the sensor signals that may have been exaggerated with increasing load and increasing slope. If this was the case, than the differences in data between conditions may be due to a difference in the magnitude of artifacts between conditions. The authors must show some data to show otherwise.
3. According to the revised manuscript, parameters were computed for the entire 2-minute intervals without considering gait cycles. Hence, the data for the different conditions may not represent multiples of full gait cycles and the number of steps will have differed between conditions based on the difference in cadence. In fact, based on the described calculation, for instance medio-lateral trunk displacement may represent just one outlier during any particular trial. The proper way of computing parameters would have been to split the data into gait cycles, calculate the parameters for each gait cycle and take the average of all steps for each condition. The discussion in section 4.2 must also be revised accordingly.
4. In Figure 8, projected ML trunk displacement was on average 0 cm for all conditions and slopes. If ML trunk displacement was calculated as the average ML trunk position across the entire trial, then this result indicates that participants walked with symmetrical trunk sway to either side. This parameter would not be influenced by greater sway to the right and left side. If the authors were interested in whether the trunk sway displacement from the center position was greater, i.e. if trunk sway was greater, then the side-to-side excursion of trunk displacement should have been calculated for each step and average across all steps for each condition.
5. Statistical analysis: It is not sufficient to report a theoretical sample size calculation. Again, the authors must provide data at least for one parameter, e.g., from work on loaded gait or work on walking on an incline.
6. The functional relevance of the observed differences between conditions must be discussed in greater detail.
Specific comments
7. Line 30: Central Nervous System – use lower case.
8. Line 95: revise – “and spatiotemporal”
9. Line 98: skeletal -> skeletally
10. Table 1: Musculoskeletal characteristics: How were side to side difference calculated? It should always be greater value minus smaller value for each participant before taking the average.
11. Line 130: Intraclass – lower case
12. Line 134: whose -> where the
13. Line 135: “The speed was…” Please revise this sentence. It reads awkward.
14. Line 140: whose -> with a
15. Line 144-: Why is some text bold here?
16. Extensor – erector: Please use consistent terminology throughout the manuscript.
17. Line 183: Please use abbreviation – it was already defined above.
18. Figure 1: post acquisition – post acquisition analysis?
19. Section 2.5: sampling trunk sway data at 1000 HZ is extremely high. The frequency of trunk movement during walking is expected to be around 1 Hz. Please specify if and how the signals were filtered. Please also provide more information on how AP and ML displacement of trunk sway was calculated. See also general remark above.
20. Line 240: EMG recordings -> EMG parameters?
21. Line 280: “the slope increased it was higher in the FL…” This is unclear. What slope do the authors refer to?
22. Line 294: “…was similar but in general greater compared to…” Was this statistically significant? If not, then revise to reflect the results of the statistical analyses.
23. Line 308: “…it was smaller during uphill…” Was this statistically significant? If not, then revise to reflect the results of the statistical analyses.
24. Figures: Please indicate the results of the posthoc tests. I.e. Figure 6: Was the difference between conditions only significant for the 0% slope? If yes, then this should be indicated accordingly. Was there a main effect or not? Please also report if there was no significant interaction.
25. Line 319: “AP displacement of trunk sway” -> “AP trunk displacement” Please revise throughout the entire manuscript and in figures, also for ML trunk displacement.
26. Line 322: Please also report if there was no significant interaction.
27. Table 3: Please provide a unit for heart rate and add information on the perceived exertion scale with anchors in the Table footnotes.
28. Line 533: It is unclear if the authors refer to their own data or to the cited articles. Why is the walking speed listed here 4 km/h?
29. Line 540: The authors did not measure kinematic parameters and should refrain from discussing these.
Author Response
Dear Editor, Dear Reviewer
I would like to thank you for the opportunity you are giving us to resubmit the article with the corrections requested (highlighted in the manuscript in yellow). The entire manuscript was scrutinized and all comments were responded to as adequately as possible to comply with the reviewer’s comments. Several parts in the Results section as well as other parts of the manuscript (highlighted in the manuscript in grey) were altered following re-calculation and analysis of data. However, we are willing to respond to further comments if any of them have not been understood or adequately answered.
________________________________________________________________________________________________
Reviewer’s comment 1: As stated in the first review, the walking speed of 5 km/h uphill with load carriage is very fast. Although the authors added the aspect of fatigue in their limitations, the relevance of data at this exertion state is still questionable as most people would not choose to perform at this exertion level. This aspect must be included in the discussion section. The total duration of the experiment was 40 minutes. Although the authors randomized the order of conditions, there is a substantial fatigue effect that may explain the large variability in data among participants,
and
Reviewer’s comment 6: The functional relevance of the observed differences between conditions must be discussed in greater detail.
Author’s responses for comments 1 and 6 collectively: The possible implications of fatigue induced by walking on level and sloped surfaces with the same, relatively fast speed, and the functional relevance of the observed differences between conditions were discussed in a separate paragraph in the Discussion section titled “Practical implications” (Please see Discussion, Lines 582-636).
Reviewer’s comment 2: The information regarding the avoidance of interference of the IMU and the EMG electrodes with the front/backpack is not sufficient. Considering the muscles assessed, the backpack must have been placed on top of the sensors. Presumably, this placement will lead to artifacts in the sensor signals that may have been exaggerated with increasing load and increasing slope. If this was the case, then the differences in data between conditions may be due to a difference in the magnitude of artifacts between conditions. The authors must show some data to show otherwise.
Authors’ response for comment 2: Six subjects were tested twice, with and without contact of the IMU and EMG electrodes with the front/backpack, to investigate the effect of possible interferences on the signals. Contact between the IMU and EMG electrodes with the front/backpack was prevented by placing specially shaped hard cardboard around both the IMU and the EMG electrodes.
|
a |
b |
c |
d |
e |
Figure 1a-e. Locations of specially shaped hard cardboards placed around both the IMU (a-b) and the EMG electrodes located on the thoracic and lumbar erector spinae (c-d) as well as on rectus abdominis (e) to prevent contact with the carried load.
The potential interference of (i) the backpack with the IMU and the EMG electrodes placed on the thoracic and lumbar erector spinae during BL walking and (ii) the front pack with the EMG electrodes placed on the rectus abdominis during FL walking, were examined at all slopes under investigation. The statistical differences between IMU/EMG data acquired with and without contact of the sensors with the front or the backpack were investigated using paired t-test.
Non-contact of the IMU and the EMG electrodes with the front back or the backpack did not significantly affect the recording of the signals. Furthermore, both trunk sway and muscle activity demonstrated the same pattern of increase as the treadmill's slope increased with participants walking with both contact and non-contact of the IMU and EMG electrodes with the front back or the backpack, respectively.
|
Please see the figure in the attached file a |
Please see the figure in the attached file b |
Please see the figure in the attached file c |
Figure 2a-c: Graphs depicting trunk sway-based data such as the 95% EA (a), and the projected AP (b) and ML trunk displacement (c) acquired with and without contact of the IMU sensor with the backpack during level and uphill BL walking.
|
Please see the figure in the attached file a |
Please see the figure in the attached file b |
Please see the figure in the attached file c |
|
Figure 3a-c: Graphs depicting the TES (a) and LES (b) the EMG activity recorded during level and uphill BL walking as well as the RAB (c) activity recorded during level and uphill FL walking. |
||
The results of this pilot study was briefly mentioned in “Testing procedure” (Please see Lines 154-158) and presented in full as Supplementary material (please see Appendix I).
Reviewer’s comment 3: According to the revised manuscript, parameters were computed for the entire 2-minute intervals without considering gait cycles. Hence, the data for the different conditions may not represent multiples of full gait cycles and the number of steps will have differed between conditions based on the difference in cadence. In fact, based on the described calculation, for instance medio-lateral trunk displacement may represent just one outlier during any trial. The proper way of computing parameters would have been to split the data into gait cycles, calculate the parameters for each gait cycle and take the average of all steps for each condition. The discussion in section 4.2 must also be revised accordingly.
and
Reviewer’s comment 4: In Figure 8, projected ML trunk displacement was on average 0 cm for all conditions and slopes. If ML trunk displacement was calculated as the average ML trunk position across the entire trial, then this result indicates that participants walked with symmetrical trunk sway to either side. This parameter would not be influenced by greater sway to the right and left side. If the authors were interested in whether the trunk sway displacement from the center position was greater, i.e., if trunk sway was greater, then the side-to-side excursion of trunk displacement should have been calculated for each step and average across all steps for each condition.
and
Reviewer’s comment 19: Section 2.5: sampling trunk sway data at 1000 HZ is extremely high. The frequency of trunk movement during walking is expected to be around 1 Hz. Please specify if and how the signals were filtered. Please also provide more information on how AP and ML displacement of trunk sway was calculated. See also general remark above.
Authors’ response for comments 3, 4, and 19 collectively: We would like to thank you for the clarification in the comment about normalizing the data according to the gait cycle. As mentioned in the authors' responses to the previous review, the IMU was synchronized with the OCR gait analysis system, and thus trunk sway and spatiotemporal parameters of gait were provided for each step separately by the system’s application software. Therefore, the mean of each of these parameters corresponding to each step performed in the intermediate 120 sec of walking at each slope and carrying mode ​​divided by the number of steps, not the average of the two minutes of recording the aforementioned parameters in each condition, was used in the statistical analysis. As for the calculation of the projected AP and ML trunk displacements, both were recalculated based on the absolute value recorded in each slope and load-carrying mode. Similarly, the average EMG activity recorded at each step performed in the intermediate 120 seconds of walking at each slope and load carrying mode divided by the number of steps, not the average EMG activity recorded for 120 seconds presented in the previous version of the manuscript, were calculated for consistency with the values obtained for trunk sway and spatiotemporal parameters of gait. This was achieved by developing a computer program written in Python 3.5. A full description of the detection of the steps performed is presented in the “Testing procedure” (Please see Lines 190-199).
Furthermore, IMU data acquisition was preset by the manufacturer at 1000 Hz, with no option to change. Regarding filtering (it was assumed that the reviewer's comment was about filtering the data for analysis) no filter was used considering that all values recorded corresponded to true trunk sways.
Reviewer’s comment 5: Statistical analysis: It is not sufficient to report a theoretical sample size calculation. Again, the authors must provide data at least for one parameter, e.g., from work on loaded gait or work on walking on an incline.
Authors’ response to comment 5: The authors wish to maintain the initial approach in calculating the sample size for the reasons presented in the previous response letter.
Specific comments
Reviewer’s comment 7: Line 30: Central Nervous System – use lowercase.
Authors’ response to comment 7: Central Nervous System was written using lowercase (Please see Lines 32-33).
Reviewer’s comment 8: Line 95: revise – “and spatiotemporal”
Authors’ response to comment 8: Revision was made as suggested by the reviewer (Please see Line 98).
Reviewer’s comment 9: Line 98: skeletal -> skeletally
Authors’ response to comment 9: “…skeletal…” was replaced with “…skeletally…” (Please see Line 102).
Reviewer’s comment 10: Table 1: Musculoskeletal characteristics: How was side to side difference calculated? It should always be greater value minus smaller value for each participant before taking the average.
Authors’ response to comment 10: Following the reviewer’s suggestion, side-to-side differences were re-calculated and highlighted in Table 1 based on the absolute value for the related variables i.e., shoulder protraction asymmetry, leg length discrepancy, pelvic rotation, and pelvic tilt (Please see Table 1 in Page 3).
Reviewer’s comment 11: Line 130: Intraclass – lowercase
Authors’ response to comment 11: “…Intraclass…” was written using lowercase (Please see Line 135).
Reviewer’s comment 12: Line 134: whose -> where the
Authors’ response to comment 12: “…whose…” was replaced with “…where the…” (Please see Line 139).
Reviewer’s comment 13: Line 135: “The speed was…” Please revise this sentence. It reads awkward.
Authors’ response to comment 13: Revision was made as suggested by the reviewer (Please see Lines 140-143).
Reviewer’s comment 14: Line 140: whose -> with a
Authors’ response to comment 14: “…whose…” was replaced with “…with a…” (Please see Line 146).
Reviewer’s comment 15: Line 144-: Why is some text bold here?
Authors’ response to comment 15: text in bold was removed
Reviewer’s comment 16: Extensor – erector: Please use consistent terminology throughout the manuscript.
Authors’ response to comment 16: “…erector spinae…” instead of “…trunk extensor…” was used throughout manuscript
Reviewer’s comment 17: Line 183: Please use abbreviation – it was already defined above.
Authors’ response to comment 17: The abbreviation for MIVC was used as suggested by the reviewer (Please see Line 202).
Reviewer’s comment 18. Figure 1: post acquisition – post acquisition analysis?
Authors’ response to comment 18: “…post acquisition…” was replaced with “post-acquisition analysis” as suggested by the reviewer. (Please see Figure 1 in Page 5).
Reviewer’s comment 20: Line 240: EMG recordings -> EMG parameters?
Authors’ response to comment 20: “…EMG recordings…” was replaced by “…EMG parameters…” as suggested by the reviewer (Please see Line 259).
Reviewer’s comment 21: Line 280: “the slope increased it was higher in the FL…” This is unclear. What slope do the authors refer to?
Authors’ response to comment 21: The slope refers to the treadmill’s slope. The text was modified accordingly (Please see Line 301).
Reviewer’s comment 22: Line 294: “…was similar but in general greater compared to…” Was this statistically significant? If not, then revise to reflect the results of the statistical analyses.
and
Reviewer’s comment 23: Line 308: “…it was smaller during uphill…” Was this statistically significant? If not, then revise to reflect the results of the statistical analyses.
and
Reviewer’s comment 26: Line 322: Please also report if there was no significant interaction.
Authors’ response to comments 22, 23 and 26 collectively: Text in the suggested sections as well as the other relevant sections of the manuscript was modified to comply with the reviewer’s suggestions and the revised statistical analysis based on the re-calculation of EMG and trunk sway data (Please see the highlighted in grey lines in the Results section of the manuscript).
Reviewer’s comment 24: Figures: Please indicate the results of the post hoc tests. i.e., Figure 6: Was the difference between conditions only significant for the 0% slope? If yes, then this should be indicated accordingly. Was there a main effect or not? Please also report if there was no significant interaction.
Authors’ response to comment 24: All figures were modified accordingly to comply with the reviewer’s suggestion.
Reviewer’s comment 25: Line 319: “AP displacement of trunk sway” -> “AP trunk displacement” Please revise throughout the entire manuscript and in figures, also for ML trunk displacement.
Authors’ response to comment 25: “…AP/ML) displacement of trunk sway…” was replaced with “…AP/ML trunk displacement…” throughout the manuscript.
Reviewer’s comment 27: Table 3: Please provide a unit for heart rate and add information on the perceived exertion scale with anchors in the Table footnotes.
Authors’ response to comment 27: A unit for heart rate was added and information on the perceived exertion scale with anchors in the Table footnotes in Table 3 as suggested by the reviewer. (Please see Table 3 in page 11).
Reviewer’s comment 28: Line 533: It is unclear if the authors refer to their own data or to the cited articles. Why is the walking speed listed here 4 km/h?
Authors’ response to comment 28: 4 km/h was the speed used by other authors (Liu et al 2020) and although it was lower compared to the speed used in our study, it was used in addition to the similar observations on step length it was among the few studies that have investigated the effects of back-loaded walking on inclined surfaces (Please see Lines 565-568).
Reviewer’s comment 29: Line 540: The authors did not measure kinematic parameters and should refrain from discussing these.
Authors’ response to comment 29: Text was modified to comply with the reviewer’s suggestion (Please see Line 574).
